# Anticancer Potential of Flavonoids: An Overview with an Emphasis on Tangeretin

**DOI:** 10.3390/ph16091229

**Published:** 2023-08-30

**Authors:** Francisco Canindé Ferreira de Luna, Wallax Augusto Silva Ferreira, Samir Mansour Moraes Casseb, Edivaldo Herculano Correa de Oliveira

**Affiliations:** 1Laboratory of Cytogenomics and Environmental Mutagenesis, Environment Section (SEAMB), Evandro Chagas Institute (IEC), BR 316, KM 7, s/n, Levilândia, Ananindeua 67030-000, Brazil; wallaxaugusto@gmail.com (W.A.S.F.); ehco@ufpa.br (E.H.C.d.O.); 2Oncology Research Center, Federal University of Pará, Belém 66073-000, Brazil; samircasseb@ufpa.br; 3Faculty of Natural Sciences, Institute of Exact and Natural Sciences, Federal University of Pará (UFPA), Rua Augusto Correa, 01, Belém 66075-990, Brazil

**Keywords:** flavonoids, tangeretin, cancer, anti-cancer, anti-proliferative, anti-metastatic

## Abstract

Natural compounds with pharmacological activity, flavonoids have been the subject of an exponential increase in studies in the field of scientific research focused on therapeutic purposes due to their bioactive properties, such as antioxidant, anti-inflammatory, anti-aging, antibacterial, antiviral, neuroprotective, radioprotective, and antitumor activities. The biological potential of flavonoids, added to their bioavailability, cost-effectiveness, and minimal side effects, direct them as promising cytotoxic anticancer compounds in the optimization of therapies and the search for new drugs in the treatment of cancer, since some extensively antineoplastic therapeutic approaches have become less effective due to tumor resistance to drugs commonly used in chemotherapy. In this review, we emphasize the antitumor properties of tangeretin, a flavonoid found in citrus fruits that has shown activity against some hallmarks of cancer in several types of cancerous cell lines, such as antiproliferative, apoptotic, anti-inflammatory, anti-metastatic, anti-angiogenic, antioxidant, regulatory expression of tumor-suppressor genes, and epigenetic modulation.

## 1. Introduction

Cancer cells possess biological properties that confer the ability to develop and become malignant. The spread of these cells occurs through a variety of tumor physiological strategies, including the maintenance of proliferative signaling, evasion of growth suppressor genes, evasion of immune destruction, induction of replicative immortality, activation of invasion and metastasis, promotion of angiogenesis, resistance to cell death, deregulation of cellular energy and metabolism, unlocking of phenotypic plasticity, and cellular senescence. These properties are acquired at different stages of neoplasia in diverse types of cancer. This ability is triggered by strong genomic instability caused by successive mutations of regulatory genes, the infiltration of tumor-promoting immune cells, non-mutational epigenetic reprogramming, and polymorphic microbiomes [1,2,3]. Elucidating these “hallmarks” of cancer is the subject of intense experimentation to explore cancer therapies, as the effective intervention of any of these tumor characteristics can potentially improve and refine anticancer therapeutic treatments against cancer.

Within the therapeutic approaches to various types of cancers, the resistance to multiple drugs (MDRs) exhibited by tumor cells is considered the main cause of chemotherapy effectiveness failure. This occurs due to cellular physiological responses triggered by the tumor, including the evasion of drug-induced apoptosis, activation of detoxification pathways, reduction in drug uptake, and activation of DNA repair mechanisms [4]. From this perspective, the use of natural products in clinical trials has been instrumental in suppressing resistance mechanisms, and hence of utmost importance in the search for new genotoxic therapeutic approaches against tumors. These products have enabled the development of more effective strategic combinations with fewer side effects for the treatment of various types of cancer, in addition to improving our understanding of cancer cell defense and resistance mechanisms [5]. The alteration of gene expression patterns in tumor cells is linked to genetic and epigenetic events. Aberrant epigenetic modifications through DNA methylation, nucleosome remodeling, histone modifications, and non-coding microRNAs play a crucial role in tumor initiation and uncontrolled cellular progression. The understanding and discovery of drugs capable of restoring or inhibiting these abnormal epigenetic mechanisms represent a significant advance in the means of cancer control [6,7,8,9]. In this context, natural phenolic compounds have gained prominence in anticancer pharmaceutical studies. These compounds, found in plants and fruits, are described as potent epigenetic agents that regulate DNA methylation, histone modification, and microRNAs in cancer therapy. They have shown effectiveness when combined with chemotherapy drugs or even when used in combination with other natural compounds. These promising findings have driven research and the development of new therapeutic strategies for cancer treatment [10,11]. This review article aims to provide an overview of the main anticancer properties of flavonoids demonstrated in various scientific studies, considering the prominent characteristics of cancer, with an emphasis on tangeretin.

## 2. Polyphenols

Polyphenols constitute a diverse group of phytochemicals associated with secondary metabolism in plants. They have antidiabetic, antiosteoporotic, cardioprotective, neuroprotective, antioxidant, anti-inflammatory, antimicrobial, immunomodulatory, and anticancer properties [12,13]. They protect plants from ultraviolet radiation and microbial infections, serve as signaling molecules during the pollination process, and modulate plant growth hormones [14,15,16]. Based on their structure, polyphenols are classified into non-flavonoids (curcuminoids, lignans, stilbenes, and tannins) and flavonoids [17] (Figure 1).

Flavonoids (or bioflavonoids) represent an extensive class with over 10,000 described subtypes of compounds [18,19,20]. They are the most abundant phenolic compounds in the human diet, ubiquitously found in fruits, seeds, roots, cereals, teas, and wines [21,22,23,24]. Although some are colorless, their etymology is derived from the Latin word “*flavus*,” which means yellow [25]. In addition, flavonoids exist in various derived forms, including glycosylated, acetylated, methylated, and sulfated aglycones [20,26,27].

## 3. Structure and Classification of Flavonoids

Structurally, flavonoids have fifteen carbons in their chemical structure (C6-C3-C6), consisting of two benzene rings (A and B) connected by a heterocyclic pyran ring (C) (2-phenyl-1,4-benzopyran) [28,29] (Figure 2).

The classification of flavonoids is based on the arrangement of the hydroxyl groups, the degree of unsaturation, and the oxidation of the heterocyclic C-ring. The main subclasses include flavones, flavonols, flavanones, flavanonols, flavanols, isoflavones, anthocyanidins, and chalcones [30,31,32,33,34] (Figure 3). Flavanones and flavanonols show a saturated benzopyran ring, the difference between them being the presence of a hydroxyl group on carbon number three of the benzopyran ring in flavanols. Similarly, flavanols also have a saturated benzopyran ring and hydroxyl groups on carbon number three; however, they differ in the absence of a carbonyl group on carbon number four of the benzopyran ring. Anthocyanins are hydroxylated at carbon number three and have two double bonds. Isoflavones have a double bond between carbon numbers two and three of the benzopyran ring, with the phenyl group attached to carbon number three. Flavonoids that do not have the benzopyran ring are called minor flavonoids. This is true for chalcones, characterized by the absence of the heterocyclic benzopyran ring with oxygen [35,36,37].

## 4. Antitumor Activity of Flavonoids

These bioactive compounds exhibit many biological properties, including antioxidant, antiviral, antifungal, antibacterial, anti-inflammatory, antidiabetic, anti-obesity, antimutagenic, cardioprotective, and anticancer activities [38,39,40,41,42]. Concerning their anticancer activities, the recognized importance of flavonoids has led to efforts and challenges to elucidate the molecular and cellular mechanisms of antitumor effects [43]. This awareness has been accompanied by an increasing number of scientific publications comparing the human health benefits of flavonoids in the field of oncology with those of other medical specialties, such as endocrinology, cardiology, and neurology [44]. Epidemiological studies support the chemopreventive benefits of flavonoids when included in the human diet, with their intake correlated with a lower risk of developing some tumors, such as gastric, breast, prostate, and colorectal cancers [45,46]. Flavonoids mediate anti-neoplastic mechanisms by modulating reactive oxygen species (ROS) levels in tumor cells, inhibiting carcinogens, pro-inflammatory pathways, angiogenesis, autophagy, inducing apoptosis, and inhibiting tumor proliferation and invasion [47,48,49,50,51,52,53,54,55] (Figure 4).

Even though the anticancer efficacy of flavonoids is described in the literature, the pharmacological activity of these compounds may be limited due to their water insolubility. The low solubility of flavonoids presents a double-edged sword in the therapeutic field. On one hand, their reduced absorption due to low solubility does not confer toxicity to the organism. On the other hand, it also becomes a problem as it may reduce their chemosensitizing effectiveness due to inefficient absorption [55,56].

In order to overcome this disadvantage, nanoparticle-based delivery systems have been developed aiming to improve the bioavailability and absorption of drugs in cancer therapy. These drug-carrying nanocarriers, such as polymeric micelles, liposomes, dendrimers, and carbon nanotubes, have been extensively investigated to ensure the chemotherapeutic and chemosensitizing effectiveness of drugs targeted to cancer cells [57,58]. In this context, the production of flavonoid-loaded phytoparticles has added advantages to the treatment, prevention, and clinical perspectives of cancer. These phytoparticles increase the bioavailability of compounds with low solubility, prolong the half-life of drugs, improve blood absorption, and reduce gastrointestinal degradation. Moreover, this delivery system allows for lower quantities of flavonoids to be used, thereby decreasing the risk of toxicity in non-tumor cells [59,60,61,62].

As an example, the effect of an oral nanoparticle delivery system of chitosan containing an encapsulated epigallocatechin-3-*O*-gallate (EGCG) flavonoid has been described as excellent in vitro in human melanoma cells and in vivo in melanoma tumor xenografts. It promotes cell growth inhibition and the induction of apoptosis in vivo, showing enhanced effectiveness in vitro when compared to native EGCG treatment [63]. These results stem from efforts to improve the bioavailability of EGCG based on previous research focusing on melanoma cancer, aiming to optimize the anticancer effects of antiproliferation and pro-apoptosis physiologically [64]. These findings reaffirm that the encapsulation (nanochemoprevention) of substances with chemopreventive activity in EGCG nanoparticles can be an efficient alternative in cancer treatment [65].

In the same way, treatment with EGCG nano-emulsion (nano-EGCG) in lung cancer cells showed the anti-tumor effects between EGCG and nano-EGCG groups. Both treatment groups blocked tumor cell growth. Importantly, the nano-EGCG treatment inhibited cell migration and invasion in a dose-dependent manner, achieved through the stimulation of the adenosine monophosphate-activated protein kinase (AMPK) signaling pathway [66]. This pathway is altered in the metabolic reprogramming of cancer cells and is responsible for conferring resistance to cancer-fighting drugs, preventing the autophagy of cancer cells [67,68].

Moderate levels of reactive oxygen species (ROS) resulting from mitochondrial activity act as redox signaling molecules in growth, differentiation, and cell proliferation pathways. However, excessive levels of ROS induce DNA mutations, protein and lipid damage, and stimulate pro-oncogenic signaling pathways, thus contributing to carcinogenesis [69,70]. Tumor cells have significantly higher ROS levels in the tumor microenvironment compared to the homeostatic conditions of non-tumor cells. However, excess ROS can be harmful to cancer cells, leading to cell death. Consequently, tumor cells develop adaptive detoxification mechanisms in response to excessive ROS [71,72]. As the elevation of ROS can trigger apoptosis in cancer cells, therapeutic strategies aimed at modulating ROS levels in cancer treatment have shown the efficacy of anticancer drugs [73,74,75]. In this sense, flavonoids are described to exhibit antioxidant biological activity in non-tumor cells and pro-oxidant activity by inducing increased oxidative stress in cancer cells, thereby inhibiting cell proliferation signaling, suppressing pro-inflammatory cytokines, promoting apoptosis, necrosis, and autophagy activation [28]. The ability to scavenge oxygen reactive species is related to the presence of a large number of phenolic hydroxyl groups in the molecular structure of flavonoids, where intense electron exchange facilitates substitution reactions with free radicals, forming a more stable compound. Therefore, the higher the number of hydroxyl groups, the greater the oxidant and pro-oxidant capacities of the flavonoid [76,77]. Ovarian cancer cells treated with flavonoids apigenin, luteolin, and myricetin showed an intracellular increase in ROS levels in a dose-dependent manner compared to untreated control cells, resulting in the activation of the intrinsic apoptotic pathway, cell cycle arrest, and anti-invasion [78]. Similarly, it was described that the flavonoid quercetin triggered cell death in cancer cells by positively regulating ROS levels [79]. The expression of the transglutaminase 2 (TGM2) gene is generally associated with poor prognosis in pancreatic cancer and is involved in its initiation, inflammation, and progression, making it a target marker in studies analyzing drugs with chemosensitizing activity [80,81,82]. Treatment with kaempferol suppressed pancreatic cancer growth in vivo and in vitro. It was observed that treated cells had decreased TGM2 expression, and the increase in ROS induced apoptosis through the Akt/mTOR signaling pathway [83]. The therapeutic potential of flavonoids in modulating ROS demonstrates that their pro-oxidant activity can positively contribute to anticancer research.

In order for excessive cell growth to be achieved, cancer cells reprogram their energy metabolism. This reprogramming is directly related to the maintenance and aggressiveness of neoplastic cells [84]. In this sense, glutathione is a ubiquitous endogenous antioxidant tripeptide (γ-Glu-Cys-Gly; GSH) found in eukaryotic cells, being responsible for maintaining cellular redox homeostasis by eliminating reactive oxygen species (ROS), a cellular metabolic byproduct [85,86,87]. Glutathione (GSH) metabolism has been investigated in tumor progression and explored as a targeted therapeutic strategy for cancer [87,88]. The positive modulation of GSH levels is directly related to the response to cellular detoxification mechanisms. This provides advantages to various types of cancers, as it is crucial for the elimination and detoxification of certain chemotherapeutic agents, thus conferring therapeutic resistance. Moreover, high GSH levels contribute to tumor development and increase metastasis events [89]. On the other hand, the reduction (depletion) in GSH levels leads to certain types of cell death, such as apoptosis, necroptosis, ferroptosis, and autophagy, providing a foundation for studies exploring the suppression of GSH levels in chemosensitization approaches in cancer therapies, making tumor cells prone to the cytotoxic and cytoprotective effects of antineoplastic substances [90]. In this direction, it has been observed that tangeretin is able to reduce oxidative stress in human hepatocellular carcinoma induced by *tert*-Butyl Hydroperoxide (t-BHP) by inhibiting GSH depletion in the cell [91]. Similarly, in cisplatin-induced liver lesions in rats treated with tangeretin, protective activity against cellular oxidative stress was observed, and an increase in antioxidant defense was also observed, as evidenced by elevated GSH levels [92]. Hence, this flavonoid is capable of reducing cellular stress and restoring the antioxidant defense system.

Epigenetic mechanisms are commonly associated with cancer development. In breast cancer, the expression pattern of certain tumor suppressor genes is related to methylation patterns. DNA methylation plays a critical role in controlling gene activity and nuclear architecture, being the most extensively studied epigenetic modification in humans. It is involved in the regulation of various biological processes, such as cell differentiation, embryogenesis, X-chromosome inactivation, microRNA expression, suppression of transposable elements, and genomic imprinting [93,94,95]. Hence, DNA methylation is an epigenetic mark associated with gene silencing, as it affects chromatin structure and blocks the access of binding factors, preventing the expression of the genes. This pattern can be stably maintained throughout life or undergo changes during aging [96]. Hypermethylation of CpG islands in the promoter region of tumor suppressor genes is an early event in various types of cancer. Consequently, CpG island hypermethylation in the promoter region can affect genes involved in cell control, DNA repair, apoptosis, and angiogenesis. In breast and ovarian cancers, hypermethylation is found in the promoter region of the BRCA1 gene, which acts as a tumor suppressor and is responsible for preventing the uncontrolled proliferation of cells [97,98]. Hypomethylation of DNA also triggers neoplastic transformations when it causes chromosomal instability, thus reactivating or activating oncogenes [99]. The literature highlights flavonoids as epigenetic modifiers in breast cancer. Epigallocatechin-3-gallate (EGCG), genistein, daidzein, resveratrol, and quercetin are capable of restoring the expression pattern of silenced tumor suppressor genes, such as BRCA1 and BRCA2, by inhibiting the enzymes called DNA methyltransferases (DNMTs). These enzymes are responsible for catalyzing the gene silencing process in the promoter region of the genes [99,100,101]. The restauration of the original expression patterns of these suppressor genes by the flavonoids was observed in different breast cancer cells, resulting in decreased proliferation and cancer cell migration [100]. The knowledge of the antitumor properties and ability of flavonoid subclasses (anthocyanidin—delphinidin, flavones—apigenin, luteolin, tangeretin, isoflavones—genistein, flavanones—hesperetin, silibinin, flavanol—EGCG, flavonols—quercetin, kaempferol, and fisetin) to modulate epigenetic enzymes, such as DNA methyltransferases (DNMTs), acetyltransferases (HATs), histone methyltransferases (HMTs), and histone deacetylases (HDACs), reinforce the incentive for research on therapeutic combination approaches involving these natural compounds that alter the epigenetic marks related to cancer development and progression along with drugs already used for cancer treatment [102].

The study of the mechanisms of action of apoptotic caspases in cancer has been explored through the use of antineoplastic drugs as a therapeutic strategy to overcome resistance and control the proliferation of cancer cells. The modulation of apoptosis under the action of natural products has demonstrated efficacy in inducing neoplastic cell death, representing an additional alternative to common chemotherapeutic agents employed in cancer treatment. It opens up a path for the development of new antineoplastic drugs, focusing on the apoptotic events executed by caspases [103,104,105,106,107]. The deregulation of the caspase cascade is implicated in the disruption of programmed cell death and directly related to the pathophysiology of cancer (evasion of apoptotic programming). The apoptotic imbalance resulting from negative caspase regulation is considered one of the causes of the resistance to tumor death found in cancer treatment [108,109,110]. The apoptotic proteolytic activation of caspases is executed through intrinsic (mitochondrial) and extrinsic (cytoplasmic) pathways. The intrinsic pathway is activated as feedback in response to cellular stress caused by cytotoxic substances, DNA mutations, hypoxia, cytoskeletal disruption, etc. [111,112,113]. In lung cancer cells treated with the flavonoid hesperetin, cell death by apoptosis was induced through the extrinsic pathway by increasing the expression levels of death domains genes, such as FADD, caspase-8, and FAS. The same study also mentioned that increased cell death occurred independently of the suppressor protein p53 and the pro-apoptotic protein Bax [114]. Treatment with malvidin and an analysis through flow cytometry showed that apoptotic activity was triggered by increased effector caspase-3 in myeloid and lymphoid leukemia cells in a dose-dependent manner, resulting in cell death [115]. Another study, also using flow cytometry, as well as Western blot and real-time PCR, showed the result of cell death by apoptosis in gastric cancer cells, where silibinin increased the level of caspases-3 and -9, followed by the inhibition of the transducer of signaling and activator of transcription 3 (STAT3) pathway, which is related to tumor growth and metastasis [116].

The molecular protective effect of flavonoids on DNA reduces the damage caused by carcinogens and promotes cellular genomic stability, allowing the development of strategies to treat neoplasms [19]. Table 1 presents the developed studies that describe the antitumor properties of flavonoids in various types of cancers.

## 5. Antineoplastic Activity of Tangeretin

The flavonoid tangeretin (5,6,7,8,4′-pentamethoxyflavone) is found in the peels of citrus fruits, especially oranges and tangerines. Studies have reported the beneficial bioactivities of this flavonoid, including its anti-asthmatic, antioxidant, anti-teratogenic, anti-inflammatory, neuroprotective, and anticancer properties [143,144,145,146]. Citrus flavonoids have demonstrated their potential anticarcinogenic activity both in in vivo and in vitro experiments by targeting cancer-related cellular processes, such as carcinogen bioactivation, cell signaling, cell cycle regulation, inflammation, and angiogenesis [147,148]. Tangeretin exhibits pharmacological properties, such as antiproliferative, anti-invasive, and anti-metastatic, and can induce apoptosis in specific cancers [149,150,151,152,153] (Figure 5). Experimental molecular analyses have focused on exploring and elucidating the cellular pathways involved in the metabolic activity of these flavonoids, further supporting their chemotherapeutic potential [147].

Using a proteomic approach, Yumnam et al. [154] investigated the effect of tangeretin on a human gastric cancer cell line. They observed that the treatment inhibited the activity of markers (*PKCs, MAPK4, PI4K, PARP14*) associated with poor prognosis in various cancers related to cell migration, proliferation, chemoresistance, the suppression of apoptosis, and differentiation. Remarkably, this study also sheds light on the importance of the PKC family as a novel biomarker in gastric cancer, as the overexpression of one of its members, PKCε, known for its anti-apoptotic functions, was inhibited by tangeretin treatment, which ultimately induced apoptosis in the gastric cell line. These findings highlight the potential of the PKC family as a promising marker and therapeutic target for treating gastric cancer with tangeretin.

Gliomas are responsible for originating the majority of brain tumors, presenting a high mortality rate, infiltrative growth, and low early detection. Despite intense conventional therapeutic advancements in gliomas, a cure for these tumors is still considered distant [155]. Increasing clinical data and research demonstrate that natural compounds emerge as promising agents in therapies aimed at combating GBM [156]. The potential antineoplastic effect of tangeretin was demonstrated by inducing cell cycle arrest and cell death in GBM. Tangeretin treatment positively modulated the expression of the *PTEN* gene and cell cycle regulating genes, and induced cell cycle arrest in G2/M and apoptosis. This suggests that tangeretin can be used as a chemopreventive agent in treating GBM. This assay reinforces the importance of further studies on the antitumor activity of tangeretin in nervous system tumors [157].

In in vivo experiments conducted on rat mammary carcinogenesis, tangeretin exhibited promising results. After cancer induction by 7,12-dimethylbenz(α)anthracene, oral treatment with this flavonoid affected markers associated with uncontrolled cell growth (PCNA, COX-2, and Ki-67). It effectively arrested the division of tumor cells at the G1/S phase by positively regulating the p53/p21 genes. Additionally, tangeretin demonstrated remarkable antimetastatic and antiangiogenic activities by inhibiting matrix metalloproteinases (MMPs) MMP-2/MMP-9 and the vascular endothelial growth factor (VEGF), respectively [158]. Sangavi and Langeswaran, using in silico approaches [159], investigated the inhibitory effect of natural compounds on liver cancer, targeting cyclooxygenase 2 (COX-2), an enzyme associated with inflammatory and carcinogenic processes (angiogenesis, metastasis, and apoptosis resistance). They found that tangeretin exhibited efficacy, showing a favorable pharmacokinetic profile for absorption, distribution, metabolism, excretion, and toxicity. These properties are essential for synthesizing new antineoplastic drugs and confirming the antitumor activity of this compound on the target cyclooxygenase 2 (COX-2) in hepatocellular carcinoma (HCC). The suppression of cyclooxygenase 2 (COX-2) was also observed in the epidermal cells of mice exposed to ultraviolet-B radiation (UVB). This occurred by blocking mitogen-activated protein kinase (MAPK) signaling and NF-kB activation and inhibiting the increase in ROS levels in cells upon UVB exposure, providing cellular protection against oxidative stress. These results suggest that the anti-inflammatory and modulatory effects of tangeretin may have a chemopreventive effect on skin cancer [160].

In macrophage cells, the process of inflammation induced by lipopolysaccharide (LPS) triggered a substantial increase in pro-inflammatory cytokines (IL-1, IL-6, and TNF-α) that were activated by the messenger molecule nitric oxide (NO). After incubation with tangeretin, the activation of anti-inflammatory cytokines (IL-4, IL-13, TNF-β, and IL-10) was observed, along with significant inhibition of inducible nitric oxide synthase (iNOS) and COX-2 [161]. As LPS is responsible for promoting inflammation and cell migration in certain types of cancer [162,163,164,165], the control of cellular inflammation described by the action of tangeretin may contribute to cancer treatment.

Antineoplastic agents administered as part of therapies can induce apoptosis in cancer cells. However, these agents often cause cytotoxicity in noncancerous cells, particularly immature immune system cells (myelocytes) and leukocytes (lymphocytes). In this context, the use of tangeretin in human leukemic cells (HL-60) from promyelocytic leukemia inhibited their growth by inducing apoptosis without promoting cytotoxicity or other side effects in immune system cells [128]. Combining antineoplastic agents (synergistic therapy) has resulted in more effective therapeutic strategies and the mitigation of side effects associated with chemotherapeutic agents commonly used for cancer treatment. When tangeretin was combined with the synthetic 5-fluorouracil (5-FU) and administered in treating certain solid tumors, significant antitumor activity was observed in colon cancer cells. This co-exposure decreased the antioxidant levels in tumor cells, resulting in oxidative stress through the accumulation of reactive oxygen species (ROS), triggering a DNA damage response and directing the cells toward apoptosis via c-Jun N-terminal kinases (JNKs). Significantly, tangeretin synergistically intensified the induction of apoptosis by 5-FU. Similarly, the co-treatment also caused a decrease in mitochondrial activity [166].

Another well-known chemotherapeutic agent commonly used to treat various human cancers is cisplatin, or *cis*-diamindichloroplatin (II). Cisplatin proves its effectiveness by causing DNA damage in tumor cells, leading to apoptosis. However, the side effects, such as kidney problems, weakened immunity, gastrointestinal problems, bleeding, and hearing damage, limit its applicability and effectiveness [167,168]. The use of tangeretin in acute liver injury caused by cisplatin in rats showed a protective effect against these histopathological deformations, underscoring its effect on one of the severe side effects of cisplatin treatment. Moreover, tangeretin reduced inflammatory mechanisms by neutralizing tumor necrosis factor-alpha (TNF-α) and stimulating interleukin-10 (IL-10) [92].

The expression of cell division-retarding tumor suppressor proteins, such as p21, p53, and p27, was increased in colorectal carcinoma cells when treated with tangeretin, thus promoting the inhibition of cell growth by triggering the blocking of enzymes responsible for regulating cell cycle progression and cyclin-dependent kinases (CDK2) and (CDK4) [125]. The elevation of tumor suppressor protein levels, followed by the inhibition of CDK, shows important anticancer effects, preventing neoplastic cells from entering division and ensuring the evasion of the suppression mechanism targeted against carcinogenesis. Considering the anticancer activities exhibited by citrus flavonoids, a study of the effects of a synthetic derivative of tangeretin (5,4’-didemethyltangeretin (PMF2)) in human prostate cancer cells demonstrated the restoration of P21 gene expression through epigenetic mechanisms of demethylation, followed by the blocking of DNMT 3B and HDACs protein expressions, thereby inhibiting cell proliferation [167].

Given the properties demonstrated for the different tumor characteristics in various types of cancer, tangeretin presents itself as a promising agent in the development of anticancer therapeutic strategies.

**Figure 5 pharmaceuticals-16-01229-f005:**
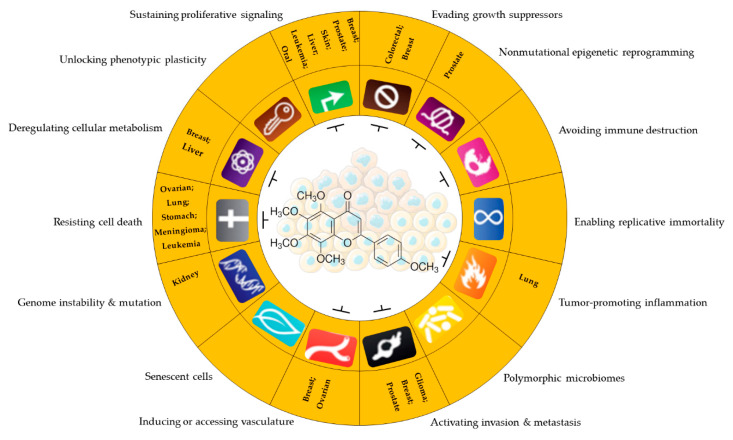
Antitumor potential of tangeretin. Anticancer activity of tangeretin under some cancer characteristics promotes uncontrolled cell progression and resistance to therapies in different types of tumors. Sustaining proliferative signaling: [127,152,160,161,167,168]; evading growth suppressors: [127,167]; nonmutational epigenetic reprogramming: [167]; tumor-promoting inflammation: [161,169]; activating invasion and metastasis: [151,170,171]; inducing or accessing vasculature: [170,172]; genome instability and mutation: [143]; resisting cell death: [128,129,173]; deregulating cellular metabolism: [92,174]. Parts of the figure are drawn by using pictures from Servier Medical Art. Servier Medical Art by Servier is licensed under a Creative Commons Attribution 3.0 Unported License (https://creativecommons.org/licenses/by/3.0/ accessed on 12 August 2023).

## 6. Conclusions and Future Perspectives

The biological potential, bioavailability, cost-effectiveness, and minimal side effects of flavonoids position them as promising cytotoxic anticancer compounds in the optimization of therapies and in the search for new drugs for the treatment of cancer. However, it is crucial to address the challenges that limit the effectiveness of flavonoids in the field of oncology, including pharmacokinetics (low solubility and stability, interaction with intestinal microflora, and metabolic interaction with receptors), pharmacodynamics, epidemiological studies (long duration, delays in data collection and categorization, absence of participant data, and exposure to heterogeneous factors), and isolation/purification of their natural sources. Among citrus flavonoids, tangeretin exhibited antitumor activities against cell proliferation. In addition, it also synergistically promoted improvements in reducing the side effects and yield when combined with some traditional chemotherapy drugs already implemented in cancer treatments. In order to provide more robust scientific knowledge about the antineoplastic activity of flavonoids, further studies are needed to examine the dosage, bioavailability, efficacy, and safety to establish the clinical use of these promising anticancer therapeutic agents.

## Figures and Tables

**Figure 1 pharmaceuticals-16-01229-f001:**
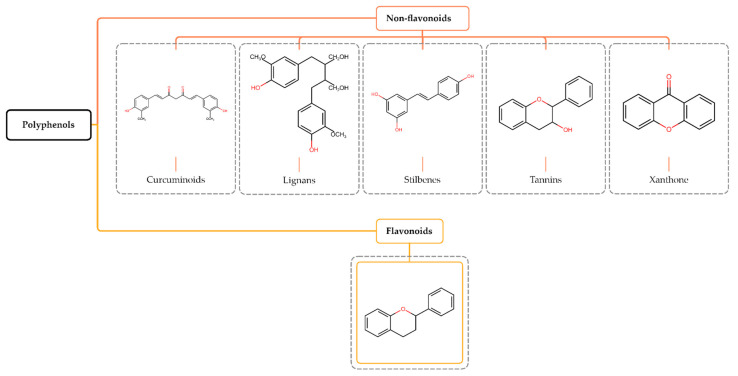
Structure and classification of polyphenols. Polyphenols are phytochemical compounds found in plants, fruits, and natural compounds, divided into non-flavonoids and flavonoids.

**Figure 2 pharmaceuticals-16-01229-f002:**
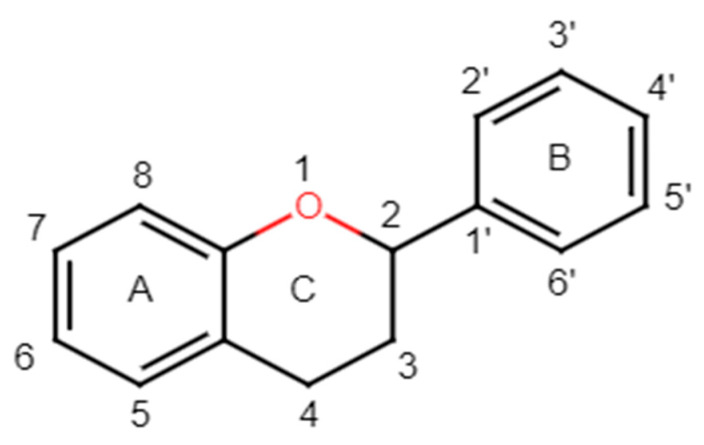
Main molecular structure of flavonoids.

**Figure 3 pharmaceuticals-16-01229-f003:**
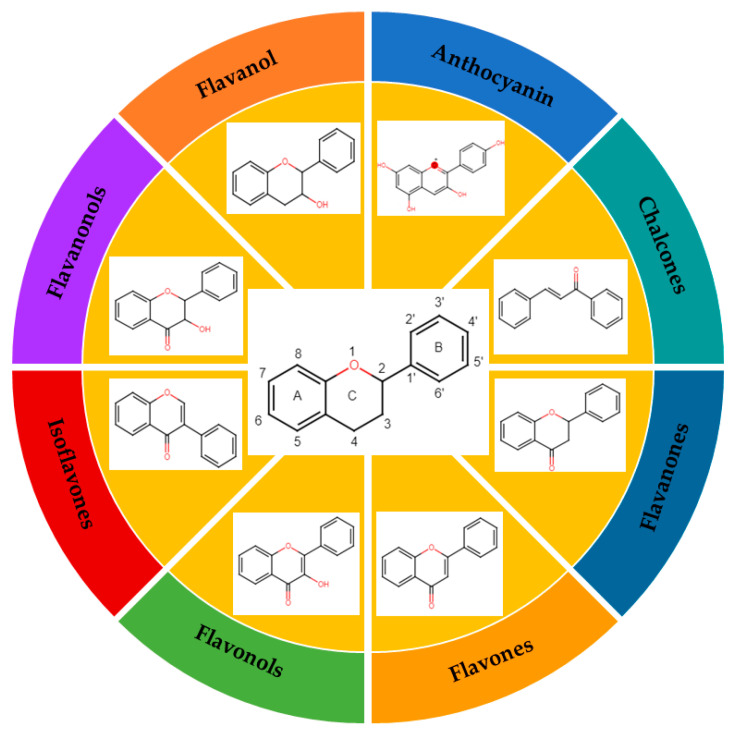
Subclassification of flavonoids.

**Figure 4 pharmaceuticals-16-01229-f004:**
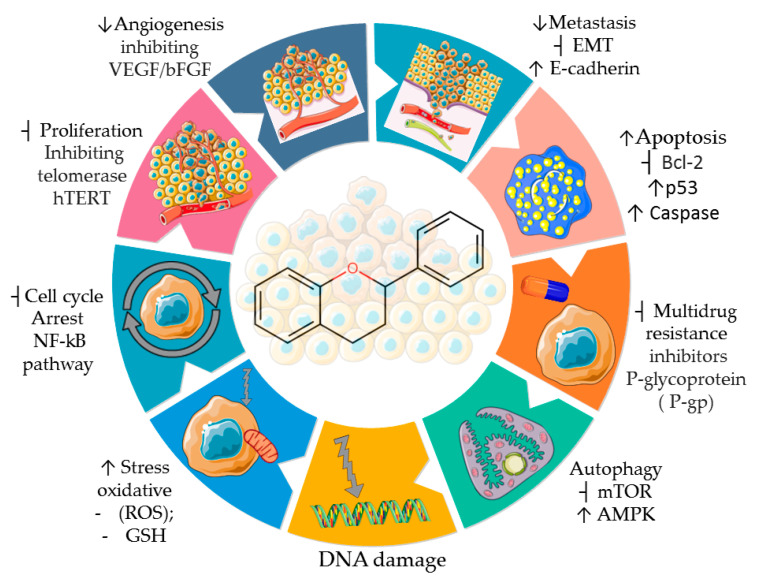
Properties and some anticancer action mechanisms of flavonoids. Parts of the figure are drawn by using pictures from Servier Medical Art. Servier Medical Art by Servier is licensed under a Creative Commons Attribution 3.0 Unported License (https://creativecommons.org/licenses/by/3.0/ accessed on 12 August 2023).

**Table 1 pharmaceuticals-16-01229-t001:** Subclasses of flavonoids and their compounds with antitumor activity described in cancer cell lines.

Subclasses	Compounds	Antitumor Activity	Cancer/Cell
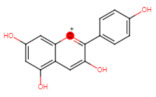 Anthocyanin	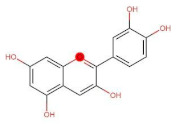 Cyanidin	Anti-proliferative Anti-metastatic Apoptosis ↓ (NF-κB) Anti-metastatic Apoptosis ↓ (NF-κB)	Kidney/ 786-O; ACHN; [117] Colorectal/ HCT116; HT29; SW620; [118]
	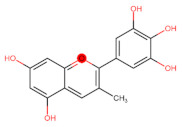 Delphinidin	Apoptosis ┤ (ERK; NF-κB)	Breast/ MDA; MB-453; BT-474; [119]
	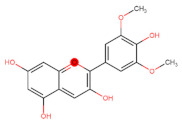 Malvidin	Anti-proliferative Apoptosis	Leukemia/ SUP-B15; KG-1; [115]
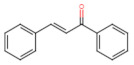 Chalcones	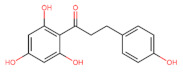 Phloretin	Anti-proliferative ┤ Migration ↑ ROS	Prostate/ PC3; DU145; [120]
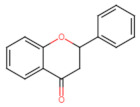 Flavanones	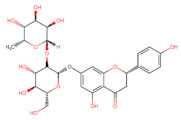 Naringin	Apoptosis ↓ (PI3K/AKT) Anti-proliferative Anti-metastatic ┤ (Zeb1) Autophagy ↓ (PI3K/AKT/mTOR)	Thyroid/ TPC-1; SW1736; [121] Osteosarcoma/ MG63, U2OS; [122] Gastric/ AGS; [123]
	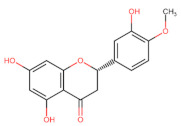 Hesperidin	Apoptosis ↑ (FADD/caspase-8)	Lung/ H522; [114]
	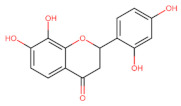 Eriodictyol	Anti-proliferative Apoptosis ┤ mTOR/PI3K/Akt Anti-proliferative Apoptosis -Anti-metastatic ┤ PI3K/Akt/NF-κB	Lung/ A549; [124] Glioma/ U87MG; CHG-5; [125]
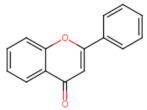 Flavones	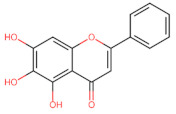 Baicalein	Induction apoptosis Autophagy ┤ (PI3K/AKT)	Breast/ MCF-7; MDA-MB-231; [126]
	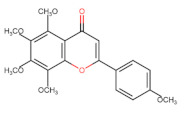 Tangeretin	Anti-proliferative ┤ (Cdk2/Cdk4) Anti-proliferative Apoptosis ↓ (MMP) ↑ Caspases-3, -8, -9	Colorectal/ COLO 205; [127] Leukemia/ HL-60 [128] Gastric/ AGS; [129]
	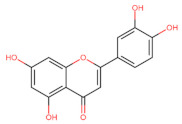 Luteolin	↑ p53 Apoptosis ┤ DNA metiltransferas	Colo/ HT-29 [130]
	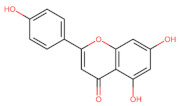 Apegenin	Apoptosis ↑ BAX, CYT c ,SMAC/DIABLO, HTRA2/OMI, CASP-3 and -9	Leukemia/ THP-1; Jukart; [131]
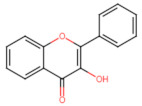 Flavonols	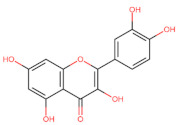 Quercetin	Antioxidant ↓ (ROS)	Ovarian/ C13* cisplatin-resistant (C13*) [132]
	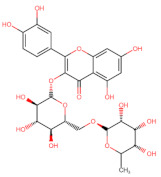 Rutin	Induction apoptosis ↑ (caspases-3, -8, -9)	Colo/ HT-29 [133]
	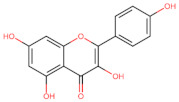 Kaempferol	Apoptosis ┤ Akt/mTOR	Pancreas/ PANC-1; Mia PaCa-2; [83]
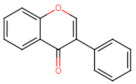 Isoflavones	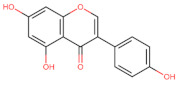 Genistein	Anti-proliferative Cell cycle arrest G2/M	Breast/ MCF-7; ERβ1; MDA-MB-231/ERβ1; [134]
	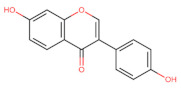 Daidzein	Anti-proliferative ┤ NF-κB	Lung/ A594 e 95D; [135]
	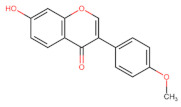 Formononetin	Anti-proliferative ┤ EGFR-Akt	Lung/ HCC827; H3255; H1975; A549; H1299; [136]
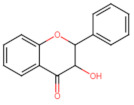 Flavanonols	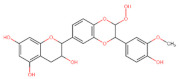 Silibinin	Anti-proliferative Induction apoptosis Cell cycle arrest G2/M ┤ (STAT3)	Gastric/ MGC803; [116]
	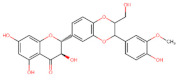 Sylimarin	Anti-proliferative Apoptosis ↑ (Caspases-5, -8)	Oral/ HSC-4; YD15; Ca9.22; [137]
	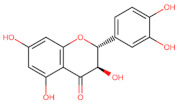 Taxifolin	Anti-proliferative ┤ (EMT) ↑ E-cadherin Cytotoxicity Cell cycle arrest G2/M	Lung/ A549; H1975; [138] Colorectal/ HCT116; HT29; [139]
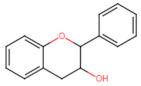 Flavanol	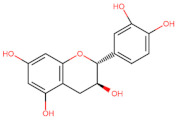 Catechin	Anti-metastatic ┤ (Wnt)	Breast/ MCF-7; HTB-26; Pancreas/ PANC-1; AsPC-1 Colorectal/ HT-29; Caco-2; [140]
	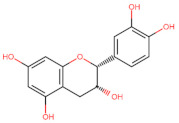 Epicatechin	Apoptosis ↑ (DR4/DR5)	Breast/ MDA-MB-231; MCF-7; [141]
	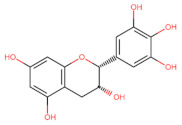 Epigallocatechin (EGCG)	Anti-proliferative Anti-metastatic ↑ AMPK Anti-proliferative Apoptosis Cell cycle arrest S ┤ EGFR/RAS/RAF/MEK/ERK	H1299, A549; [66] Thyroid/ TT; TPC-1; ARO; [142]

## Data Availability

Data sharing not applicable.

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
