# Peer review of "Anticancer Potential of Flavonoids: An Overview with an Emphasis on Tangeretin"

_pharmaceuticals, 2023, doi:10.3390/ph16091229_

Round 1

Reviewer 1 Report

The current MS entitled ``therapeutic properties of flavonoids in cancer and anti-tumor effectiveness of tangeretin`` describes the potential of flavonoids as promising anticancer compounds. This work cannot be published in its current form the following issue should be addressed.

Major issues

A- English needs careful revision through the whole MS. There are many typing and grammatical mistakes.

B- The MS is poorly written and lacks proper organization of the topics to be covered.

C- The current work lacks significant novelty. There are many reported works discussing the same work you can find them under this link: https://scholar.google.com/scholar?hl=en&as_sdt=0%2C5&q=THERAPEUTIC+PROPERTIES+OF+FLAVONOIDS+IN+CANCER&btnG=;

https://scholar.google.com/scholar?hl=en&as_sdt=0%2C5&q=FLAVONOIDS+%2B+CANCER+%2B+review&btnG=

D- The previous review on this topic should be discussed and the significance of this work over the previously published work should be clarified.

E- Research methodology is not mentioned. The period that this work cover should be included.

F- Inclusion and exclusion criteria for the selection of the reported articles in this work should be mentioned.

G- There is no need to have separate section for `` Phenolic Compounds`` because all the data mentioned are well-known. This can be added in the introduction.

H- All chemical structures should be redrawn, considering unified bond length and the font size in all structures.

I- All structures present in figure 1, do not represent the basic skeleton for the class of natural compounds stated below it. Particularly for the non-flavonoids. Why the authors mentioned `` Polyphenols are phytochemical compounds found in plants, fruits, and natural compounds, divided into non-flavonoids and flavonoids. `` in figure1`s legend.

J- Section 3. ``Structure and Classification of Flavonoids`` should be summarized since all mentioned information are already know for flavonoids.

K- All structures should be removed from Table 1 as the basic skeleton for each flavonoids subclasses were drawn if Figure 2.

L- Figures for the mentioned flavonoids structures should be included. `` Cyanidin, Delphinidin, Malvidin, ……………

M- In table 1. Arbutin is not flavonoid, arbutin is a glycoside; a glycosylated hydroquinone.

N- There are unlimited number of flavonoids with therapeutic efficacy in cancer, why the authors mentioned only this limited number of flavonoids???? Need clarification in the manuscript.

O- Summarized illustration for all the possible mechanisms of flavonoids in cancer should be included.

P- Are there any commercially approved flavonoids in the market, or in clinical trials for cancer?. Also, are there semisynthetic derivatives of flavonoids approved for cancer?. Discuss in the MS.

Q- Section for Future perspectives and recommendations should be included.

R- Why the authors emphasized on Tangeretin, this should be clarified.

Minor issues

Title

1- Authors should follow the Journal guidelines in the writing style of the title.

2- The title is misleading and confusing, what is the purpose for including ``tangeretin`` in the title. Therefore, the title should be modified to be targeted on the focus of this work.

Abstract

3- In the abstract ``tangerine`` is a type of citrus fruit not a flavonoid. Check through the whole MS.

4- The abstract needs rewriting to clearly reflect the purposes of this work. There is no relation between what is the major part covered in the body of the MS and what is present in the abstract `` In this review, we emphasize the 20 properties of tangerine, a flavonoid from citrus fruits that has shown antitumor activity against some cancer markers related to cell proliferation and cancer cell resistance.``

5- The abstract should highlight what is the work reported in the MS. The abstract seems like an introduction about the flavonoids. Some of the findings should be written in the abstract.

Keywords

There are many keywords that look the same. Instead of flavonoids; flavones; cancer; anti-cancer; anti-proliferative;……………) that do not reflect what is present in the abstract.

 Introduction

1- introduction lacks sufficient references.

2- More statistical information about the incidence of cancer should be included.

Conclusion

Conclusion is the same as abstract. It should be modified.

Remove references from conclusion.

English needs careful revision through the whole MS. There are many typing and grammatical mistakes.

Reviewer 2 Report

Nice work. However, I have some general and specific suggestions for authors.

General comments

1. Please unify terminology related to polyphenols. It is not quite same as phenolic compounds. The second one is wider term gathering also simple phenolic compounds like phenolic acids. However, since target here are flavonoids I think that better choice, in this case, is term polyphenols. And, please never it can be polyphenolic. Please check a whole document and unify.

2. Technical issue- please check the title. I think it should not be written with all capital letters?

3. In Figure 1 I think that outer term should be "Phenolic compounds" since you include phenolic acids.

Specific comments

All are listed below with an appropriate Line number(s) from text in order to facilitate tracking.

Line 44: typo - split word "Under" from previous sentence with space.

Line 66: As I explained, terms phenolic compounds and polyphenols are not completely the same. Please correct.

Line 94: Check here- three times mentioned flavanols but some must be wrong and incorrect.

Lines 98 and 100: typos- at the end of sentence not .. but .

Line 109: Suggest to be ".... importance of these bioactive compounds intake". Delete "consuming" it is not appropriate here since no one will eat this compounds per se.

Line 139: In chemistry letter O in name of glycosides must be always written in Italic. Please correct. Also, from here further in text please all "in vitro" and "in vivo" terms put in Italic. Check all and correct.

Line 204: Same chemical comments for "tert" as for letter O in the Line 139. Correct.

- On Page 8 in the Table 1 check name for "Apigenin"? It is given now as "Apegenin".

- On Page 9 in the Table 1 check name for "quercetin"? It is given now as "Quercertin".

Line 283: Why in plural here "these flavonoids"? Aren't you talking about one compound- Tangeretin here?

Line 329: I do not understand what are these "tangeretin fractions"? How one compound can have fractions? Please elaborate/explain/correct.

Line 350: Same chemical comments for "cis" as for "tert" and letter O. Please correct.

Line 371: "tangeretin" not "tangerine" here, right? Same typo in the Lines 374 and 394. Please check/correct.

- Please check references. They are not all in the Line with journal policy. For instance, in references 127 and 161 surnames of authors are given with all capital letters.

Kind regards.

Reviewer 3 Report

This short review presents the anticancer properties of flavonoids and, on this background, the anti-tumor effectivity of tangeretin. The review is generally well written and may be useful for other researchers.

Remarks

Phenolic compounds. The classification of phenolic compounds presented in the manuscript is proper and generally used. However, to be precise, it can be reminded that some phenolic acids, such as ferulic acid or vanillic acid have only one phenol group and thus are not “polyphenols”.

Glutathione: It is better to introduce this compound as γ-Glu-Cys-Gly to emphasize the atypical bond between Glu and Cys. It is worth mentioning that GSH is a substrate for glutathione peroxidase and glutathione S-transferase; the latter reaction is important for the conjugation and subsequent removal of xenobiotics, including cancer chemotherapeutics.

Table 1: “apigenin”, apigenin?, “quercertin”, quercetin?

Line 371, 394 and Fig. 4: Do the authors mean tangerine or tangeretin? Apart from tangeretin, tangerins contain several other flavones such as nobiletin and sinensetin, so ascribing all the effects of tangerines to tangeretin does not seem to be fully justified.

Round 2

Reviewer 1 Report

The quality of the work has been improved.

English needs refinement.